# Phase 3 Randomized, Multicenter, Placebo-Controlled Study to Evaluate Safety, Immunogenicity, and Lot-to-Lot Consistency of an Adjuvanted Cell Culture-Derived, H5N1 Subunit Influenza Virus Vaccine in Healthy Adult Subjects

**DOI:** 10.3390/vaccines10040497

**Published:** 2022-03-23

**Authors:** James Peterson, Esther Van Twuijver, Eve Versage, Matthew Hohenboken

**Affiliations:** 1J. Lewis Research, Inc., Salt Lake City, UT 84109, USA; jpeterson@foothillfamilyclinic.com; 2Clinical Development, Seqirus, Inc., 1105 BJ Amsterdam, The Netherlands; esther.vantwuijver@seqirus.com; 3Clinical Development, Seqirus, Inc., Cambridge, MA 02139, USA; eve.versage@seqirus.com

**Keywords:** influenza, vaccine, cell culture-derived, H5N1, pandemic, adult, elderly

## Abstract

A cell-based process may be better suited for vaccine production during a highly pathogenic avian influenza (HPAI) pandemic. This was a phase 3, randomized, controlled, observer-blind, multicenter study evaluated safety, immunogenicity, and lot-to-lot consistency of two doses of a MF59-adjuvanted, H5N1 influenza pandemic vaccine manufactured on a cell culture platform (aH5N1c) in 3196 healthy adult subjects, stratified into two age groups: 18 to <65 and ≥65 years. Immunogenicity was measured using hemagglutination inhibition (HI) titers. HI antibody responses increased after the first aH5N1c vaccine dose, and 3 weeks after the second vaccination (Day 43), age-appropriate US Center for Biologics Evaluation and Research (CBER) and former European Medicines Authority Committee for Medicinal Products for Human Use (EMA CHMP) immunogenicity criteria were met. Six months after the first vaccination, HI titers were above baseline but no longer met CBER and CHMP criteria. No relevant changes over time were seen in placebo subjects. Solicited AEs were more frequent in the active treatment than the placebo group, primarily due to injection site pain. No serious adverse events (SAEs) related to aH5N1c- were reported. aH5N1c influenza vaccine elicited high levels of antibodies following two vaccinations administered 21 days apart and met both CBER and former CHMP immunogenicity criteria at Day 43 among both younger and older adults with a clinically acceptable safety profile. Consistency of the three consecutive aH5N1c vaccine lots was demonstrated (NCT02839330).

## 1. Introduction

In the last century, the rise in international trade and travel has increased the probability of worldwide pandemics, as seen most recently with the infectious disease, COVID-19. The primary prophylactic measure against pandemic influenza are vaccines, and the ability to rapidly develop and produce a specific monovalent vaccine targeted to a new circulating virus strain is vital to pandemic preparedness plans worldwide [1].

During the most recent influenza pandemic—due to the (H1N1)pdm09 virus or “swine flu”—an estimated 60.8 million swine flu cases with 274,304 hospitalizations and 12,469 deaths occurred between 2009 and 2010 in the US alone, and it is estimated that the swine flu caused over 500,000 deaths worldwide. However, this pandemic appeared to be less severe than would have been expected with an associated mortality rate of only 0.001% to 0.007% in the first year, whereas for other influenza pandemics the worldwide mortality rate has ranged from 0.03% for the 1968 H3N2 pandemic to 1% to 3% during the 1918 H1N1 pandemic. In addition, the 2009 H1N1 pandemic primarily affected the young and middle aged, whereas many older adults were found to have antibodies to this virus from an earlier H1N1 infection [2].

The H5N1 avian influenza virus represents another pandemic threat. In 1997, the first outbreak of highly pathogenic H5N1 avian influenza occurred in Asia (Hong Kong), which led to 18 human cases and 6 deaths before public health authorities ordered the slaughter of poultry throughout Hong Kong to stop the spread of this virus [3]. It re-emerged in 2003, leading to worldwide concerns over the possibility of an H5N1 pandemic. According to the World Health Organization (WHO), 862 human cases of H5N1 infection were identified from 2003 to 2021 and resulted in 455 deaths, representing a case fatality rate of 53% [4].

In addition to vaccine subtype, pandemic preparedness planning must consider the capacity and efficiency of the manufacturing process. Influenza vaccine manufacturing has relied on embryonated chicken eggs to produce antigens for over 50 years. During a highly pathogenic avian influenza outbreak, both egg quantity and quality may be compromised, yet rapid production of a vaccine specific against an emerging pandemic influenza strain is critical to controlling its spread.

In alignment with the US Department of Health and Human Services (DHHS) Pandemic Preparedness Plan [5], an MF59-adjuvanted cell culture-derived monovalent H5N1 pandemic influenza vaccine (aH5N1c) vaccine was developed by Seqirus, Inc. Cell culture-derived vaccines are not subject to the potential limitations of egg-based production (e.g., the need for large quantities of fertilized eggs; the potential for egg-adaption of seed virus and antigenic mismatch) and help address the medical need for safe and effective pandemic vaccines [1,6].

Previous clinical experience suggests that two doses of nonadjuvanted H5N1 influenza vaccine with 90 μg of strain-specific hemagglutinin (HA)—which represents six times the normal 15 μg/dose required for the interpandemic seasonal influenza vaccine—are necessary to induce a substantial increase in antibody responses in unprimed, immunologically naïve individuals [7]. The use of an adjuvant, however, allows a reduction in the quantity of antigen per dose (“antigen sparing”) and would potentially lead to increased vaccine production capacities [8]. In addition, the observation of enhanced and broader, i.e., cross-reactive, immune responses after vaccination with MF59-adjuvanted H5N1, and seasonal (FLUAD) vaccines is of great interest for the development of pre-pandemic vaccines, as stockpiled vaccines may be used during the early days of a pandemic before the strain matched vaccine becomes available [9,10].

To address the threat of an HPAI outbreak, e.g., H5N1, when both egg quantity and quality may be compromised, an alternative to traditional egg supply is needed. In preparation for a future H5N1 pandemic, this study evaluated the immunogenicity, lot-to-lot consistency, and safety of three consecutively produced lots of the aH5N1c pandemic vaccine in healthy subjects ≥18 years of age.

## 2. Materials and Methods

### 2.1. Study Design and Randomization

This phase 3, multicenter, randomized, observer-blind, controlled study involved subjects aged ≥18 years who were stratified into two equal age groups, 18 to <65 and ≥65 years of age, and then randomized in a 1:1:1:1 ratio to receive one of three consecutively produced aH5N1c vaccine lots (Groups A–C) or placebo (saline). Subjects received two doses of vaccine or placebo intramuscularly given three weeks apart on Day 1 and Day 22. After each vaccination, subjects remained under medical supervision at the study site for at least 30 min to observe any immediate adverse events (AEs). After the second vaccination, subjects were monitored for 12 months for safety, for a total study duration of approximately 13 months per subject.

This trial was designed, implemented, and reported in accordance with the International Conference on Harmonization (ICH), Harmonized Tripartite Guidelines for Good Clinical Practice (GCP), with applicable local regulations, and with the ethical principles laid down in the Declaration of Helsinki. An independent institutional review board approved the study protocol and informed consent form. All study subjects provided written, informed consent. The study is registered at https://clinicaltrials.gov/, accessed on 18 March 2022 (NCT02839330).

### 2.2. Study Vaccine Administration

The vaccine used for this study was an MF59-adjuvanted, cell culture-derived, monovalent, inactivated, H5N1 subunit influenza virus (A/turkey/Turkey/1/2005 NIBRG-23 strain; Seqirus Inc., Holly Springs, NC, USA). The three lots included Lot No.181053 (Group A, Lot 1), Lot No.181054 (Group B, Lot 2), and Lot No.181675 (Group C, Lot 3). Each dose was 0.5 mL and contained 7.5 μg hemagglutinin with 0.25 mL MF59. A fourth group received 0.5 mL placebo (0.9% NaCL, 2 mL vial, West-Ward Pharmaceuticals, Cherry Hill, NJ, USA) batch number 035385. Vaccines were administered on Day 1 and Day 22 as single intramuscular injections in the nondominant arm by designated site staff who did not participate in any assessment of outcomes. The subjects, investigators, and site personnel who evaluated AEs remained blinded to treatment group assignment.

### 2.3. Study Participants

The study enrolled 3196 healthy subjects ≥18 years of age who gave consent and were willing and able to comply with protocol requirements. Individuals were excluded if they had impaired immune systems, previous influenza vaccination within 7 days of starting the study or any other vaccination within 28 days of study start, or a history of H5N1 influenza or H5N1 influenza vaccination. Female subjects of childbearing potential were excluded if they were pregnant, breastfeeding, or not using adequate birth control.

### 2.4. Study Objectives and Endpoints

The coprimary study objectives were to determine lot-to-lot consistency across three consecutively produced lots of the aH5N1c vaccine in terms of geometric mean titers (GMT) and achievement of US Center for Biologics Evaluation and Research (CBER) criteria for the percentage of subjects achieving an hemagglutination inhibition (HI) antibody titer ≥1:40 [11]. Secondary immunogenicity objectives were to evaluate immune responses to the aH5N1c vaccine according to immunogenicity criteria defined by European Medicines Authority Committee for Medicinal Products for Human Use (EMA CHMP) recommendations (as applicable at the time of study conduct) 3 weeks after the second vaccine administration (Day 43) and by CBER and CHMP recommendations 3 weeks after the first vaccine administration (Day 22), as well as to evaluate immune responses to the aH5N1c vaccine 6 months after the first vaccine administration (Day 183) [11,12].

Immunogenicity endpoints were assessed by HI assay against the H5N1 vaccine strain according to standard methods, expressed as GMTs on Days 1, 22, 43, and 183 and geometric mean ratios (GMRs; Day 22/Day 1, Day 43/Day 1, and Day 183/Day1) in each treatment group (pooled aH5N1 or placebo) in the total population and by age cohort (18 to <65 years of age and ≥65 years of age). The proportions of subjects with HI ≥1:40 on Days 1, 22, 43, and 183 and those achieving seroconversion (defined as HI titer ≥1:40 for subjects negative at baseline (HI titer <1:10) or a minimum 4-fold increase in HI titer for subjects positive at baseline (HI titer ≥1:10)) on Day 22 and Day 43 were also determined for each treatment group (pooled aH5N1c or placebo) in the total population and by age cohort.

Safety endpoints included solicited local and systemic adverse events (AEs) collected on subject diary cards for 7 consecutive days after each injection. Solicited local AEs included injection site induration, erythema, ecchymosis, and pain. Solicited systemic AEs included nausea, generalized myalgia, generalized arthralgia, headache, fatigue, chills, loss of appetite, malaise, and fever (≥38.0 °C). Erythema, induration, and ecchymosis were graded as Grade 0 (<25 mm) or any (25–50 mm (Grade I), 51–100 mm (Grade II), >100 mm (Grade III). Injection site pain, systemic AEs except fever, and all unsolicited AEs were graded as mild (transient with no limitation in normal daily activity), moderate (some limitation in normal daily activity), or severe (unable to perform normal daily activity) as assessed by the investigator. Body temperature ≥39 °C was considered severe fever.

All unsolicited AEs were collected from first vaccination through Day 43. Serious adverse events (SAEs), AEs of special interest, new onset of chronic disease, AEs leading to vaccine/study withdrawal, medically attended AEs, associated concomitant medications for any of these events, and all vaccinations, were collected throughout the study. The causal relationships of AEs to the study vaccines were assessed by the investigators as either not related, possibly related, or probably related.

### 2.5. Statistical Methods

The full analysis set (FAS) included all subjects who received at least one dose of study vaccination and provided at least one evaluable serum sample at both pre- and post-vaccination timepoints. The primary analysis population was the per protocol set (PPS), which included all subjects in the FAS who received the correct vaccine to which the subject was randomized at the scheduled time points and who were not excluded due to a major protocol deviation or other reasons (e.g., withdrew informed consent). The solicited safety set included all subjects who received a study vaccination and who underwent any assessment of local and systemic site reaction and/or assessment of any use of analgesics or antipyretics. The unsolicited safety set included all subjects who received a study vaccine.

Based on data from previous studies in similar populations, a single equivalence test based on 718 subjects per lot group was determined to have a power of 95% with alpha of 0.025. Taking a dropout rate of approximately 10% into account, a total study enrollment of 3192 subjects (798 subjects per lot) was planned.

Lot consistency was assessed by determining the geometric mean titer ratio (GMT ratio) of HI antibody responses to the H5N1 vaccine strain in healthy adults three weeks after the second vaccine administration (Day 43). Lot-to-lot consistency was demonstrated if the 2-sided 95% confidence intervals (CIs) of all three pairwise GMT ratio comparisons (Group A/Group B, Group A/Group C, Group C/Group B) fell within the equivalence range of 0.667 to 1.5. Adjusted estimates of GMT ratios and their associated 95% CIs at Day 43 were computed using analysis of covariance (ANCOVA) on the log-transformed titers at Day 43 with factors for vaccine lot group, age group, center, and a covariate for the effect defined by the log-transformed prevaccination antibody titer (Day 1).

After confirmation of lot-to-lot consistency, the results of all vaccine recipients were pooled to evaluate immune responses to the aH5N1c vaccine according to the CBER criteria for HI antibody titer ≥1:40 on Day 43 as measured by age cohort and by strain-specific HI assay. For subjects aged 18 to <65 years, CBER criteria were met if the lower bound of the adjusted 2-sided 95% CI for the percentage of subjects achieving an HI antibody titer ≥1:40 was ≥70%. For subjects ≥65 years, CBER criteria were fulfilled if the lower bound of the adjusted 2-sided 95% CI for the percentage of subjects achieving an HI antibody titer ≥1:40 was ≥60%. Adjusted proportions and 95% CI were calculated using the log-linear model with the factors for treatment and center.

Secondary immunogenicity endpoints were based on the age-appropriate CBER and CHMP criteria on Days 22, 43, and 183. The age-appropriate CBER criteria for seroconversion require the lower bound of the 2-sided 95% CI for seroconversion rate to be ≥40% or ≥30% for subjects 18–65 and ≥65 years of age, respectively. Analysis of CHMP criteria requires point estimates for seroconversion rate to be >40% and >30%, for percentage of subjects achieving an HI antibody titer ≥1:40 to be >70% and >60%, and for GMR to be >2.5 and 2.0, for subjects aged 18–60 and ≥61 years, respectively.

## 3. Results

### 3.1. Study Population

The study was conducted at 26 centers in the US between 11 July 2016 and 4 October 2017, and enrolled a total of 3196 adults, including 1597 subjects aged 18 to <65 years and 1599 subjects aged ≥65 years. Of the total enrolled population, 2394 subjects received aH5N1c and 797 received a placebo; 2234 and 747 subjects, respectively, completed the study; and 2249 and 739 were included in the PPS (Figure 1). Demography and other baseline characteristics were similar across the treatment groups (Table 1). The mean age was 58 years, and the proportion of subjects in each age subgroup was evenly distributed across the four treatment groups. The majority of participants were women, and most subjects were white. Exposure to seasonal vaccine in the previous 12 months was higher in the older age cohorts than in the younger age cohorts, consistent with typical clinical practice in the US.

### 3.2. Coprimary Objectives: Lot-to-Lot Consistency and CBER Criteria

GMTs at Day 43 for each of the aH5N1c lots were 128.6 (95% CI 118.9 to 139.1), 127.4 (117.6 to 138.0) and 132.2 (122.1 to 143.1). Pairwise comparisons of the GMT ratios demonstrated lot-to-lot consistency (Figure 2a). The CBER immunogenicity criteria were also met, with the lower bound of the 95% CI for the proportion of patients with HI ≥1:40 on Day 43 well above 70% in subjects younger than 65 years and above 60% in subjects aged ≥65 years (Figure 2b).

### 3.3. Immunogenicity

As shown in Table 2, baseline GMTs were slightly higher in older adults (≥65 years). GMTs increased from baseline in the active treatment groups at Day 22, three weeks after the first vaccination, with a further increase at Day 43, three weeks after the second vaccination, in both age groups. Increases, as assessed by GMRs, were larger in younger (18 to <65 years) than in the older (≥65 years) adults, as would be anticipated due to immunosenescence. CHMP criteria were met in both age groups on Day 43 (Appendix A). In subgroup analyses, no clinically significant differences between genders were observed.

Seroconversion rates were consistently higher among subjects receiving aH5N1c than the placebo. On Day 43, 79.9% (95% CI 77.4 to 82.3) of subjects 18–65 years of age and 54.0% (95% CI 51.0 to 57.0) of subjects ≥65 years of age receiving aH5N1c had achieved seroconversion and met the age-appropriate CBER criteria for seroconversion rates (Table 3). CHMP criteria for seroconversion were also met on Days 22 and 43 for subjects aged 18 to <60 years and on Day 43 among those aged ≥60 years in the active vaccine group (Appendix A).

### 3.4. Safety

The frequency of any solicited local or systemic AE was comparable between the aH5N1c groups and higher in the active treatment groups than in the placebo group. The proportion of subjects for whom any solicited AE was reported was lower after the second than after the first vaccination in both the active treatment and the placebo groups (Figure 3). In a subgroup analysis by gender, no clinically significant differences in safety endpoints were observed.

Injection site pain was the most common solicited local AE, reported by 49.9% of subjects who received aH5N1c compared to 14.7% of those who received the placebo. Pain was reported more frequently among younger than older subjects: 64.1% vs. 35.9% among those aged 18 to <65 and ≥65 years, respectively, in the aH5N1c treatment groups and 19.9% vs. 9.6%, respectively, in the placebo group. The majority of pain reported was of mild or moderate intensity and mostly resolved within a couple of days following vaccination. The frequency of severe pain after any vaccination was low: 4 out of 2352 (0.2%) subjects in the aH5N1c group compared to 1 out of 784 (0.1%) subjects in the placebo group. The frequency of other solicited local AEs was too low for meaningful comparison between age groups. The most common solicited systemic AE was fatigue, which was reported by 22.2% of subjects in the aH5N1c group compared to 20.4% in the placebo group. In the aH5N1c treatment group, more subjects aged 18 to <65 years reported fatigue (24.8%) than subjects aged ≥65 years (19.7%); the frequency of fatigue in the placebo group was 21.4% and 19.4% among the younger and older age groups, respectively. Malaise, headache, and myalgia were also reported more frequently by subjects aged 18 to <65 years than ≥65 years in the aH5N1c treatment groups. In both groups, solicited systemic AEs were predominantly mild or moderate in severity and mostly occurred within 3 days of injection. The frequency of severe solicited AEs was 1.9% in the aH5N1c group compared with 2.8% in the placebo group.

The proportion of subjects reporting unsolicited AEs was similar among those receiving aH5N1c (53.1%) or the placebo (52.3%) throughout the study (Figure 4). The majority of the reported unsolicited AEs were of mild or moderate intensity. No differences in frequency, severity, or nature of unsolicited AEs in the aH5N1c group compared to the placebo group were observed.

None of the serious AEs or AEs of special interest reported by subjects who received aH5N1c were considered vaccine related. Two subjects in the placebo group reported a related AE of special interest (immune thrombocytopenic purpura and polymyalgia rheumatic); these events were also considered serious AEs. During the study, 12 (0.4%) subjects had serious AEs with a fatal outcome, none of which were attributed to the study treatment, and most (n = 11) occurred after Day 43 during the follow-up period in subjects ≥65 years with underlying severe comorbidities and multiple concomitant medications.

## 4. Discussion

The results from this study demonstrated that the aH5N1c vaccine was highly immunogenic for both younger (18 to <65 years) and older (≥65 years) adults and elicited high HI titers in both age groups. The coprimary immunogenicity objectives were met, showing consistency between the three consecutively produced lots of aH5N1c and also demonstrating that the vaccine met age group-specific CBER licensure criteria for the proportion of subjects with HI ≥1:40 and for seroconversion on Day 43. Moreover, at Day 43, all three CHMP age group criteria (GMR, proportion of subjects with HI ≥1:40, and seroconversion rates) were met. GMTs, GMRs, and seroconversion rates demonstrated significantly greater antibody responses among aH5N1c recipients than placebo recipients at all time points and across age groups. In the gender-based subgroup analyses, we observed no clinically significant differences in either immunogenicity or safety endpoints.

After the first vaccination, immune responses increased from baseline, but two doses of aH5N1c augmented the response to levels that met licensing criteria. A decline in antibody titers 6 months after immunization was observed. This result was contrary to expectation based on phase 2 study results but consistent with other H5N1 studies [13,14,15,16]. Persistence of immune response was nevertheless evident, with HI titers that remained elevated over baseline 6 months after the first vaccination. Of note, in the pediatric population, both the cell and egg culture-derived MF59 adjuvanted H5N1 vaccines have been shown to elicit high antibody titers, which persisted up to 1 year after vaccination [17,18].

Although the immune response was lower in older than younger subjects, as may be expected due to age-related immunosenescence [19], both CBER and CHMP immunogenicity criteria were met. Immunosenescence encompasses a range of alterations in the immune response, including impaired function of antigen presenting cells and decreases in the number of T cells available to respond to new antigens, antibody response, high-affinity antibodies, and metabolic activity within memory CD4+ cells [20,21,22]. The MF59 adjuvant is a proprietary squalene-based, oil-in-water emulsion that improves the magnitude, breadth, and persistence of the immune response by enhancing antigen uptake at the injection site [23,24,25,26]. In multiple clinical trials, MF59-adjuvanted seasonal influenza vaccine boosted the immune response in older adults relative to standard influenza vaccines [27].

Baseline HI titers, prior to vaccination, were slightly elevated across all treatment groups. These titers appeared to be highest in the ≥65 years age group (whether receiving aH5N1c or placebo); similar results have been found by other investigators [14,28,29]. One of the possible explanations for this phenomenon is that elderly people with prolonged natural exposure to seasonal influenza viruses and/or multiple lifetime vaccinations may develop antibodies with antigenic cross-reactivity with H5N1 strains [28,29].

The aH5N1c vaccine was safe, well tolerated, and shown to have an acceptable risk–benefit profile overall. The majority of AEs were mild or moderate in severity and of a transient nature. Solicited AEs were more common with aH5N1c than the placebo, which is consistent with previous studies on the H5N1 vaccine [7]. The frequency of AEs was lower after the second than after the first vaccination, and generally, the incidence of solicited AEs was higher among subjects aged <65 than ≥65 years. The difference between aH5N1c and placebo in solicited local AEs, i.e., injection site pain, is consistent with trials of adjuvanted seasonal influenza vaccines [30,31]. The frequencies, nature, and severity of solicited systemic and unsolicited AEs were similar between the active vaccine and placebo groups, as reported in a trial of the nonadjuvanted H5N1 vaccine [7].

Cell-based production of pandemic vaccines may offer several advantages over egg-based methods. First is the potential concern of using egg-based production to combat an avian influenza pandemic. Second, egg-adaption of seed virus introduces the potential for antigenic mismatch between the vaccine and circulating strain [32,33,34,35]. In contrast, a vaccine production platform based on mammalian cell culture ensures a closer match between the original candidate virus and the vaccine virus [36]. Cell-based manufacturing may also facilitate more rapid production to meet the needs of a population beset by a pandemic [1,6].

Conclusions from this study are limited because there was no evaluation of cross-reactive antibodies, although this was assessed in a previous phase 2 study with an MF59 adjuvanted egg culture-derived H5N1 vaccine [15]. The size the study population was adequate to evaluate general safety but was not large enough to detect rare events.

## 5. Conclusions

Both coprimary immunogenicity objectives of this study were met for the aH5N1c vaccine. The ratio of GMTs for HI antibody responses to the H5N1 pandemic vaccine strain three weeks after the second vaccine administration demonstrated consistency in three consecutively produced lots of the aH5N1c vaccine. In addition, the age-appropriate CBER immunogenicity criteria for the percentage of subjects achieving an HI antibody titer ≥1:40 and those achieving seroconversion at Day 43 were met in both age groups (18 to <65 years and ≥65 years), and all three CHMP criteria (GMR, proportion of subjects with HI ≥1:40, and seroconversion rates) were met for subjects 18 to <60 years and ≥60 years of age. Vaccination with 7.5 μg of the aH5N1c vaccine elicited an immune response as shown by the increase in HI GMT after the first vaccination (measured on Day 22) that was further increased after the second vaccination (measured on Day 43). The aH5N1c influenza vaccine was well tolerated with a clinically acceptable safety profile.

## Figures and Tables

**Figure 1 vaccines-10-00497-f001:**
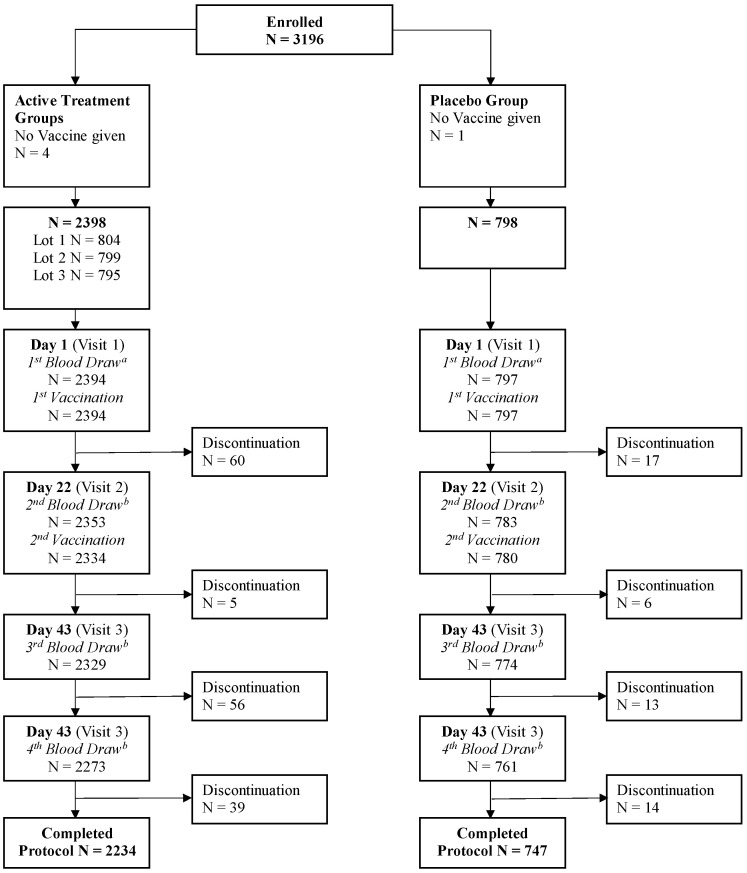
Disposition of subjects. ^a^ Includes subjects for whom blood was drawn but not on Day 1. ^b^ Includes subjects for whom blood was drawn out of the time window specified in the protocol.

**Figure 2 vaccines-10-00497-f002:**
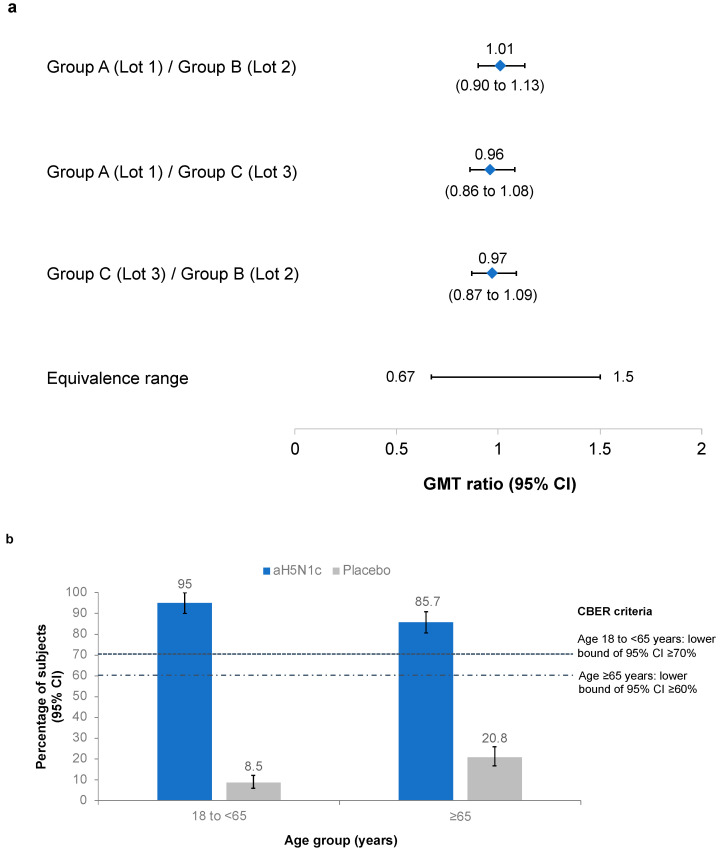
(**a**) Pairwise comparisons of the geometric mean titer (GMT) ratios of hemagglutination inhibition (HI) titers from Groups A (Lot 1, No.181053), B (Lot 2, No.181054), and C (Lot 3, No.181675). Protocol-defined equivalence range of the 95% confidence intervals (CI) was 0.667 to 1.5. (**b**) Proportion of subjects with HI ≥1:40 on Day 43 in the pooled aH5N1c and placebo groups. Center for Biologics Evaluation Research (CBER) criteria were met if the lower bound of the 95% CI was ≥70% in subjects aged 18 to <65 years and ≥60% in subjects aged ≥65 years.

**Figure 3 vaccines-10-00497-f003:**
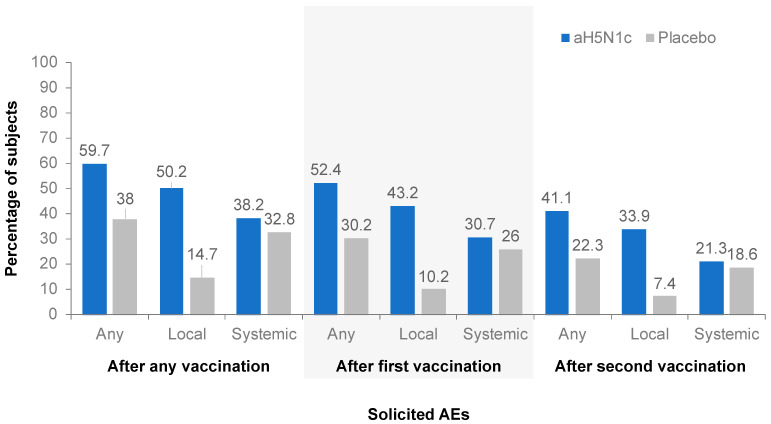
Percentages of subjects with at least one solicited adverse event (AE) within 7 days after the first, second, or any vaccination.

**Figure 4 vaccines-10-00497-f004:**
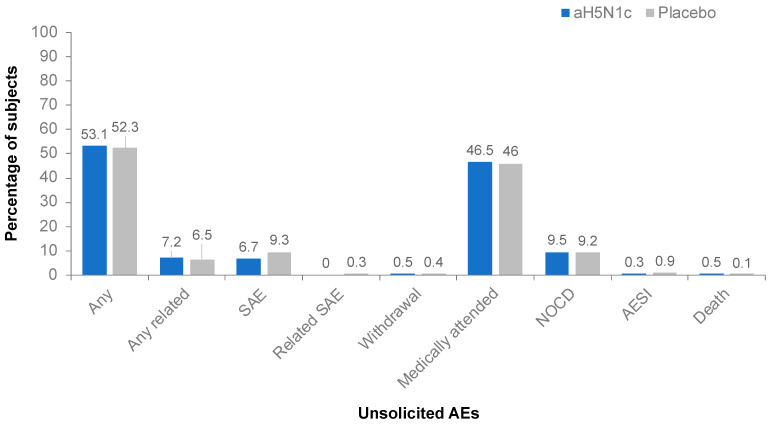
Percentages of subjects with unsolicited adverse events (AE), serious adverse events (SAE), AE leading to study withdrawal, medically attended AEs, new onset chronic disease (NOCD), AE of special interest (AESI), or death from Day 1 through study termination.

**Table 1 vaccines-10-00497-t001:** Baseline demographics and characteristics in the enrolled population.

	aH5N1c ^a^					
	Group A (n = 804)	Group B (n = 799)	Group C (n = 795)	All aH5N1c (n = 2398)	Placebo (n = 798)	Total (N = 3196)
Mean age ± SD, years	58.1 ± 17.67	57.5 ± 17.83	57.5 ± 18.24	57.7 ± 17.91	57.7 ± 18.29	57.7 ± 18.00
Age group, n (%)						
18 to <65 years	403 (50.1)	399 (49.9)	397 (49.9)	1199 (50.0)	398 (49.9)	1597 (50.0)
≥65 years	401 (49.9)	400 (50.1)	398 (50.1)	1199 (50.0)	400 (50.1)	1599 (50.0)
Female, n (%)	444 (55.2)	434 (54.3)	447 (56.2)	1325 (55.3)	438 (54.9)	1763 (55.2)
Race, n (%)						
White	668 (83.1)	679 (85.0)	674 (84.8)	2021 (84.3)	665 (83.3)	2686 (84.0)
Black	110 (13.7)	102 (12.8)	104 (13.1)	316 (13.2)	112 (14.0)	428 (13.4)
Asian	12 (1.5)	7 (0.9)	9 (1.1)	28 (1.2)	7 (0.9)	35 (1.1)
Native American or Alaskan Native	6 (0.7)	4 (0.5)	5 (0.6)	15 (0.6)	3 (0.4)	18 (0.6)
Native Hawaiian or other Pacific Islander	3 (0.4)	2 (0.3)	1 (0.1)	6 (0.3)	4 (0.5)	10 (0.3)
Other	5 (0.6)	5 (0.6)	2 (0.3)	12 (0.5)	7 (0.9)	19 (0.6)
Hispanic ethnicity, n (%)	53 (6.6)	61 (7.6)	64 (8.1)	178 (7.4)	55 (6.9)	233 (7.3)
Mean weight ± SD, kg	78.86 ± 15.0	80.22 ± 15.2	78.86 ± 15.3	79.31 ± 15.2	79.84 ± 15.3	79.44 ± 15.2
Mean BMI ± SD, kg/m^2^	27.40 ± 4.2	27.86 ± 4.1	27.41± 4.2	27.56 ± 4.2	27.60 ± 4.2	27.57 ± 4.2
Previous influenza vaccine in past 12 months, n (%)						
All subjects	430 (53.5)	416 (52.1)	426 (53.6)	1272 (53.0)	422 (52.9)	1694 (53.0)
Age 18 to <65 years	145 (36.0)	137 (34.3)	139 (35.0)	421 (35.1)	123 (30.9)	544 (34.1)
Age ≥65 years	285 (71.1)	279 (69.8)	287 (72.1)	851 (71.0)	299 (74.8)	1150 (71.9)

Abbreviations: BMI, body mass index; SD, standard deviation. ^a^ Group A = Lot 1, No.181053; Group B = Lot 2, No.181054; Group C = Lot 3, No.181675.

**Table 2 vaccines-10-00497-t002:** HI antibody responses (GMT and GMR) against H5N1 at Day 1, Day 22, Day 43, and Day 183 by age group.

	18 to <65 Years		≥65 Years		Total Population	
	aH5N1c	Placebo	aH5N1c	Placebo	aH5N1c	Placebo
Day 1, n	1116	372	1133	367	2249	739
GMT (95% Cl)	13.5 (12.8–14.2)	13.7 (12.5- 15.0)	20.5 (19.4–21.8)	20.6 (18.6–22.7)	16.6 (16.0–17.3)	16.7 (15.6–17.9)
Day 22, n	1115	370	1130	366	2245	736
GMT (95% Cl)	50.6 (47.6–53.8)	11.6 (10.4–12.9)	42.4 (40.0–45.0)	14.5 (13.1–16.0)	46.4 (44.5–48.4)	13.0 (12.1–14.0)
GMR Day 22/Day 1 (95% Cl)	3.81 (3.58–4.05)	0.87 (0.79–0.97)	2.14 (2.02–2.27)	0.73 (0.66–0.81)	2.86 (2.74–2.98)	0.80 (0.74–0.86)
Day 43, n	1076	349	1080	351	2156	700
GMT (95% Cl)	170.7 (160.5–181.6)	11.0 (9.9–12.2)	97.9 (92.1–104.1)	16.7 (15.0–18.5)	130.6 (124.8–136.6)	13.7 (12.6–14.8)
GMR Day 43/Day 1 (95% Cl)	12.70 (11.94–13.51)	0.82 (0.73–0.91)	4.90 (4.61–5.20)	0.83 (0.75–0.92)	7.96 (7.61–8.33)	0.83 (0.77–0.90)
Day 183, n	1025	341	1054	346	2079	687
GMT (95% Cl)	20.4 (19.3–21.6)	6.8 (6.1–7.4)	19.3 (18.2–20.4)	8.6 (7.8–9.5)	20.0 (19.2–20.8)	7.7 (7.2–8.2)
GMR Day 183/Day 1 (95% Cl)	1.53 (1.44–1.61)	0.51 (0.46–0.56)	0.97 (0.91–1.02)	0.43 (0.39–0.47)	1.22 (1.17–1.27)	0.47 (0.44–0.50)

Abbreviations: CI, confidence interval; GMR, geometric mean ratio; GMT, geometric mean titer; HI, hemagglutination inhibition.

**Table 3 vaccines-10-00497-t003:** Percentages of subjects (95% CI) achieving seroconversion in HI titer ^a^.

	Age 18 to <65 Years		Age ≥65 Years		Total Population	
	aH5N1c	Placebo	aH5N1c	Placebo	aH5N1c	Placebo
Day 22, n	1115	370	1130	366	2245	736
Seroconversion, % (95% CI)	40.4 (37.6–43.4)	1.9 (0.8–3.9)	24.2 (21.7–26.8)	0.3 (0.0–1.5)	32.2 (30.3–34.2)	1.1 (0.5–2.1)
Day 43, n	1076	349	1080	351	2156	700
Seroconversion, % (95% CI)	79.9 (77.4–82.3)	0.3 (0.0–1.6)	54.0 (51.0–57.0)	1.7 (0.6–3.7)	66.9 (64.9–68.9)	1.0 (0.4–2.0)
Day 183, n	1025	341	1054	346	2079	687
Seroconversion, % (95% CI)	16.2 (14.0–18.6)	0.3 (0.0–1.6)	8.0 (6.4–9.8)	1.2 (0.3–2.9)	12.0 (10.7–13.5)	0.7 (0.2–1.7)

Abbreviations: CBER, Center for Biologics Evaluation and Research; CI, confidence interval; HI, hemagglutination inhibition. ^a^ Seroconversion was defined as either a prevaccination (baseline) HI titer <1:10 and postvaccination HI titer ≥1:40 or a prevaccination HI titer ≥1:10 and a ≥4-fold increase in postvaccination HI antibody titer. Boldface indicates CBER criteria for seroconversion were met, i.e., lower bound of 95% CI ≥40% for subjects younger than 65 years and ≥30% for subjects aged ≥65 years on Day 43.

## Data Availability

The data presented in this study are available in the article and supplement as described.

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
