# Peer review of "Phase 3 Randomized, Multicenter, Placebo-Controlled Study to Evaluate Safety, Immunogenicity, and Lot-to-Lot Consistency of an Adjuvanted Cell Culture-Derived, H5N1 Subunit Influenza Virus Vaccine in Healthy Adult Subjects"

_vaccines, 2022, doi:10.3390/vaccines10040497_

Round 1

Reviewer 1 Report

The author conducted a Phase 3 randomized, multicenter, placebo-controlled study to evaluate the safety, immunogenicity, and consistency of an MF59-adjuvanted H5N1 influenza pandemic vaccine (aH5N1c on a cell-culture platform. It was shown that the aH5N1c influenza vaccine elicited high levels of antibodies following two vaccinations administered 21 days apart. No SAEs were found related to aH5N1c.  The study was well designed and methods were appropriated. The results were interpreted clearly and the conclusion was promising.  It is worthy of publication. 

Before accepting it, there are a few major concerns needing to be addressed:

  1. In many diseases including influenza A, gender is a factor that affects vaccine responses. For example, women typically produce higher antibodies than men do after vaccination. Moreover, Women typically suffered more frequently from local and systemic SEs. There is clear evidence that elderly women display a greater humoral response (HI titers) against classical A (H1N1), and pandemic A (H1N1) pmd09 (Immunol. 05 July 2021. https://doi.org/10.3389/fimmu.2021.715688). However, in this study, gender is not considered as a confounding factor that might affect the results. The justification should be provided.
  2. Significance tests were not performed in evaluating the differences of HI titers between vaccinated and placebo groups or across all treatment groups. The confidence of the results such as immune responses increased from baseline, the lower immune responses in older subjects, and persistence of immune responses from three lots, etc. is weak. The observed differences from this study might be random (see Table 1-3 and Figure 1-2).

Minors:

  1. CBER, CHMP, and AE should be given the full names when they appear at first place in Abstract (front page, line 18, 20)
  2. In Figure 2. “(B)” was not shown up (page 7).

Reviewer 2 Report

In this study the authors present data on the safety and immunogenicity of an adjuvanted H5N1 vaccine. In addition lot-to-lot variation in safety and immunigenity were also tested. The study group is large enough and the study has been well conducted.  It is also well written and the conclusions are appropriate. There a certain aspects that can be further evaluated in further processing of the manuscript.

  1. It would be very interesting to see whether there were age-related differences in the adverse events and immunogenicity of the vaccine. Often younger adults show stronger inflammatory and immune responses to vaccines (e.g. influenza vaccine) and thus this relatively large cohort would enable the analysis. The age-related groups could be divided e.g. 18-29 year, 30-39 years, 40-49 years and 50-64, and 65 and older age groups. This analysis could also provide more information of potential age-related order of vaccination groups in case there would be limited availability of the vaccine in a pandemic situation. It would also be interested to see whether in younger adults e.g. one dose would already induce good ab responses.
  2. The authors could speculate the persistence (or lack of it) of abs and how this vaccine behaves in comparison to other vaccines. I recall seeing studies with other vaccines where 6 month persistence is better. This should be discussed further.

Round 2

Reviewer 1 Report

The revisions are acceptable for publication